# Spectroscopic disentanglement of the quantum states of highly excited Cu$_2$

M. Beck [1], P. Bornhauser [1], Bradley Visser[1,4], G. Knopp [1], J. A. van Bokhoven[2,3] & P. P. Radi [1]

Transition metals, characterised by their partially filled d orbitals, provide the basis for many of the most relevant processes in chemistry, biology, and physics. Embedded as single atoms or in small clusters, they give rise to exceptional optical, chemical, and magnetic properties. So far, it has proven impossible to disentangle the complex network of excited quantum states, which greatly hinders prediction and control of material properties. Here, we apply two-colour resonant four-wave mixing to quantitatively resolve the quantum states of the neutral copper dimer. This allows us to unwind the individual spectral lines by isotopic composition and rotational quantum number and reveals a rich network of bright and perturbing dark states. While this work presents a road map for the experimental study of the bonding between and with transition metal atoms, it also provides experimental reference data for prospective quantum chemical approaches on handling systems with a high density of states.

[1] Photon Science Division, Paul Scherrer Institute, 5232 Villigen, Switzerland. [2] Energy and Environment Division, Paul Scherrer Institute, 5232 Villigen, Switzerland. [3] Department of Chemistry and Applied Biosciences, ETH Zürich, 8093 Zürich, Switzerland. [4] Present address: University of Applied Sciences and Arts, Northwestern Switzerland, 5610 Windisch, Switzerland. Correspondence and requests for materials should be addressed to P.P.R. (email: peter.radi@psi.ch)

Transition metals are key elements for many applications ranging from high-efficiency white organic light-emitting diodes[1], catalysis[2] and sensors[3], to novel approaches for cancer-cell destruction[4]. The usage of transition metals is essential in many chemical reactions. However, even when the atomic structures are known, the underlying quantum chemistry is barely understood. Less is known about processes used in chemical industry, where transition metals are employed as catalysts. While catalysis yields products like fertilisers, which are indispensable to feed the current world population[5], the lack of knowledge about the active transition metal sites obstructs the rational design of new catalysts, which are necessary for the transition towards green chemistry[6].

It has been demonstrated that not only the highest activity, but also the largest adjustability of chemical and physical properties, can be found in the quantum size regime, which spans the size range between a single atom[7] and several nanometres[8]. In this regime, the partially filled d orbitals of transition metals lead to a very large number of electronic states. Therefore, perturbation-induced mixing of energetically close states is abundant.

Such vibronic coupling, which is neglected in the Born-Oppenheimer approximation, along with spin–orbit interaction combines with coherence phenomena to enhance function in chemical and biophysical systems[9], but obstructs the understanding of such systems. To reproduce these interactions by computation, multiconfiguration self-consistent field and multireference configuration interaction methods are required[10]. However, the complexity of an excited transition metal dimer in vacuum can easily exceed the size restrictions of these methods. New approaches strongly depend on experimental verification, which up until now was impossible. The densely packed spectra cannot be resolved with standard experimental methods, as the multitude of overlapping and perturbed bands prevents the assignment of individual spectral features. Here, we provide a solution to this problem by partitioning these spectra into components of defined rotational quantum states, which can be assigned easily.

Such an approach is critical for the study of transition metal dimers given the sheer number of possible transitions in even the low-energy regime and the complex interaction of individual states with one another. The method is demonstrated on a coinage metal, as the density of states at low energies remain manageable, but increases dramatically with the promotion of an electron from the d-shell. Other transition metals have open d-shell ground state configurations, leading to a great number of individual states. An extreme of this is $Ti_2$ that has almost 5000 states in just the first 1.6 eV above the ground state[11].

The copper dimer is often considered to be the simplest transition metal dimer, with just two molecular states arising from the combination of ground state $^2S$ ($3d^{10}4s$) atoms. While this has made it a preferred test molecule for numerous computational methods, recent work emphasises the role of dicopper centres in the global carbon and nitrogen cycles[12]. The molecular ground state, $\tilde{X}^1\Sigma_g^+(0_g^+)$, and the first exited state, $a^3\Sigma_u^+(1_u)$, are formed by the two electrons in the $(4s\sigma)^2$ and $(4s\sigma)^1(4s\sigma^*)^1$ molecular orbitals, respectively, and there is little interaction between the 3d shells localised on the atoms[13].

As spin–orbit interaction is non-negligible in dicopper[14], states are labelled using their total angular momentum $\Omega$ in Hund's case (c) notation, i.e., $X0_g^+$ and $a1_u$. Despite its simple ground state, the low-lying states of a copper dimer in vacuum that can be experimentally probed by visible light, were only recently reproduced by computational methods[14]. Promotion of an electron from the 3d shell leads to a much larger number of possible combinations, with 40 molecular states formed from the bonding

of a ground state atom $^2S$ with an excited $^2D$ atom. They ensue at low energies since the $^2D$ state is only $\approx 1.5$ eV above the atomic ground state. The combination of two $^2D$ atoms gives rise to 100 states, which populate the energy range up to 5 eV.

Gas-phase measurements of the electronic structure of $Cu_2$ were reviewed by Morse in ref. [15]. Since then, several new states were observed in the ultraviolet (UV)[16] and improved spectroscopic constants for the D state were reported[17]. Despite a considerable number of spectroscopic investigations, very few of the excited states have been rotationally resolved and molecular constants determined. From the recent investigations of dicopper in this laboratory, low-lying states, including the newly found A′ $1_u$ state, were rationalised by state-of-the-art MRCI computations, including Hund's case (c) coupling terms. Except for the $X0_g^+$ and $a1_u$ state, the low-energy electronic structure is dominated by states that correspond to the $^2S + ^2D$ pair of atomic limits. The discovery of the A′$1_u$ state confirms that Hund's case (c) coupling is required to account for the number of low-energy electronic states observed. In addition, it was found that the $G0_u^+$ state can be definitely assigned to a state, which would correspond asymptotically to the ground ion-pair state[14]. More recent studies determined molecular constants for the $J0_u^+$ and $I1_u$ state and disclosed their symmetry. The $J0_u^+$-$X0_g^+$ transition could be assigned to a promotion of a $3d\pi_g$ electron to the $4p\pi_u$ molecular orbital thus determining the complete configuration of the $J0_u^+$ state as $3d^{18}3d\pi_g4s\sigma_g^24p\pi_u$, a state that dissociates adiabatically to the $3d^{10}4s + 3d^94s4p$ asymptotic limit[18]. This state is strongly bound as expected from the promotion of a d-electron to a binding 4p orbital.

In this work, by exciting the copper dimer using deep-ultraviolet light, spectra are recorded in the vicinity of the $J0_u^+$ state. In this realm, the copper dimer behaves like a typical transition metal dimer possessing open-shell d-electrons: Interactions of close lying states go beyond the reach of any computation and experimental spectra are generally too complex to be assigned. The highly selective and sensitive method applied here quantitatively unravels a rich network of interacting states by assignment of individual rovibronic transitions. The complex bonding features involving ion-pair states, perturbations, and avoided crossings are accessible. Such knowledge sheds light on the electronic structure that governs the functionality of transition metals and provides reference data for advanced computations of these important molecules.

## Results

**Spectral simplification by double-resonance spectroscopy.** The $J0_u^+$ state can be reached directly from the ground state by a strong transition, which was first described by Powers et al. in ref. [19]. However, in that work the bands recorded were seen to be accompanied by unknown features. In 1991, rotationally resolved measurements by Page et al.[20] revealed a (0-0) $J0_u^+$-$X0_g^+$ band that could not be assigned, as no set of contrived molecular constants reproduced it. In light of our results, the reason for this becomes clear.

Figure 1 shows an accurate simulation of the (0-0) $J0_u^+$-$X0_g^+$ band and a Fortrat diagram, which maps the transitions to the individual rotational quantum number $J'$ of the accessed state. There, not only transitions to the bright $J0_u^+$ state, but also hypothesised transitions into three dark vibronic states are shown. These states exhibit little transition strength from the initial ground state and cannot be accessed directly. However, they gain intensity and experience a line shift where their corresponding branches intersect with the $J0_u^+$ state. The reason behind this lies in the quantum mechanical mixing of

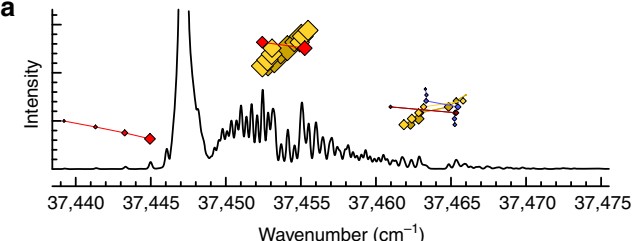

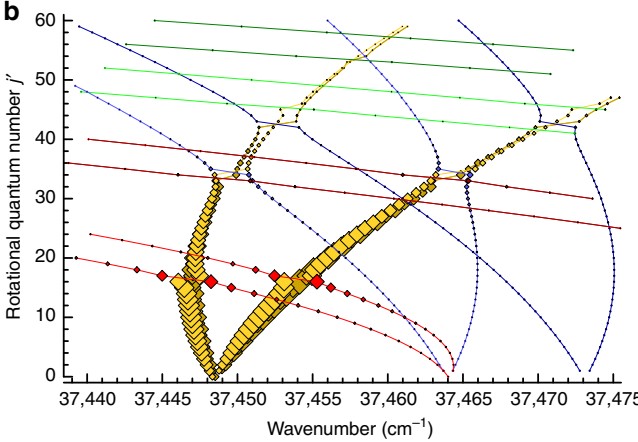

**Fig. 1** The underlying structure of a heavily perturbed band. **a** Simulation of the (0-0) $J0_u^+$-$X0_g^+$ vibronic band of the copper dimer, limited to the two most abundant isotopologues. The irregular shape of the band originates from perturbations of the electronic $J0_u^+$ state by several dark states. **b** Fortrat diagram of the same region, revealing the individual interactions occurring between the bright $J0_u^+$ state (yellow) and three dark states, $G_{62}0_u^+$ (red), $L1$ (blue), $G_{63}0_u^+$ (green), in both $^{63}Cu_2$ (dark colours) and $^{65}Cu^{63}Cu$ (light colours). While the colour denotes the vibronic state, the area of the coloured square indicates the intensity of the individual line. Band transitions are connected by lines to guide the eye. As dipole selection rules for this bands dictate that the rotational quantum number can change only by ±1 for a given initial $J''$, two branches occur for each state, a P and R-branch with $\Delta J = J' - J'' = -1$ and $\Delta J = +1$, respectively. Primes and double primes designate the excited and ground state quantum numbers, respectively

energetically close bright and dark states, given sufficient overlap of their wave functions. Consequently, both states share part of the same character and level repulsion occurs.

As there is no way to extract the data in the Fortrat diagram from an experimental one-photon spectrum (as in the one simulated in Fig. 1a), the underlying information on the molecular constants of the vibronic states and the interactions among them need to be obtained by a different experimental approach that simplifies the spectrum. In this work, we used Two-Colour Resonant Four-Wave mixing (TC-RFWM)[21] as a spectroscopic method. While this Optical-Optical Double Resonance (OODR) technique enables us to do spectroscopy on individually addressed quantum states of single isotopologues, the complex process of signal generation[22] raises substantial requirements on the experimental set-up. In particular, the target dimeric species needs to be provided with both a high and a stable number density, which was only recently achieved[18].

**Rotational disentanglement of bands**. TC-RFWM allows spectra to be collected that originate from a single known $J''$ level of the ground $X0_g^+$ state. The resulting spectra contain only one R, one P, and sometimes one Q line, depending on the symmetry of the excited state. This capability greatly simplifies the analysis of

the spectra, especially in cases where perturbations destroy the regularity of rotational levels in the upper state. When a bright state is perturbed by a dark state, the quantum mechanical mixing of these states causes their energies to shift, but also the strengths of related transitions to redistribute. Hence, the dark state becomes detectable for rotational levels where both states are sufficiently close in energy to perturb each other substantially.

Figure 2a illustrates this effect on transitions from the ground state into the vibrationally excited $J0_u^+$ ($v = 2$) state of the $^{65}Cu^{63}Cu$ dimer. Analogous to the computed Fortrat diagram in Fig. 1b, the experimental spectra are arranged as a function of the rotational quantum number $J''$ in the ground state, which was directly provided by the applied method. As a result of the ordering in the ground state, the intersections of perturbing dark states appear shifted by two rotational levels, when comparing the P-branch ($\Delta J = -1$) with the R-branch ($\Delta J = +1$). For some spectra, additional lines were visible and others remained unassigned, because of spectral overlap in the band used to determine $J''$ in the ground state. Such ambiguities can be resolved by using another transition to access the same $J'$. Figure 2b depicts the ordering and relative spacing between the rotational levels of the observed vibronic states. Most obvious is the strong interaction between the $J0_u^+$ ($v = 2$) state and the vibronic state preliminarily labelled as $G_{68}0_u^+$. The label $G_{68}0_u^+$ was chosen on purpose, as this level is part of a series of vibrational levels that was observed across all measured vibrational levels of the electronic $J0_u^+$ state. Using the isotopic shifts, it can be assumed that $G_{68}0_u^+$ corresponds to a vibrational level about $v = 68$, while its properties (vide infra) suggest an assignment to the electronic $G0_u^+$ state[15]. At its culmination, between rotational levels $J = 29$ and $J = 30$, the repulsion of these levels increases their separation from 2 to 10 cm$^{-1}$.

The perturbations between the $J0_u^+$ and $G_{xx}0_u^+$ states have been analysed by assuming a homogeneous perturbation on the basis of the observed effective rotational quantum numbers (Supplementary Table 1). Mostly, both P and R branch transitions were observed with sufficient intensity. In contrast, the vibronic state preliminarily labelled $O1$ exhibits a combination of interesting properties. Starting from low rotational levels, the dark $O1$ state just becomes visible at its intersection with the $J0_u^+$ ($v = 2$) state, where it causes a splitting of about 0.4 cm$^{-1}$ at $J = 14$. However, in this range only the P-branch is visible. From $J = 15$ onwards the situation reverses and only the R-branch is visible. Further afield of the intersection, it does not fully disappear as expected from a dark state. Finally, $O1$ approaches the intersection with $G_{68}0_u^+$ where no significant splitting is added on top of the natural separation of about 1 cm$^{-1}$ between the levels of $O1$ and $G_{68}0_u^+$, the behaviour of $O1$ changes again and we did not detect a single $O1$ line across the second intersection. Intensity anomalies between P($J + 1$) and R($J - 1$) lines have been observed and attributed to the mixing of perturbing states[23]. The effect can be explained by quantum mechanical interference that comprises information on the perturbation class. For example, parallel and perpendicular transitions (with an orbital angular momentum change $\Delta \Lambda = 0$ or ±1, respectively) display approximately equal intensities for the P($J + 1$) and R($J - 1$) lines for unperturbed states. If a perturbation occurs ($L$-uncoupling) between two states exhibiting an angular momentum difference of $\Delta \Lambda = ±1$, the relative intensity of the branches out of this state can be affected strongly. For parallel transitions the amplitude phases of P and R are identical while for perpendicular transitions they are opposite. As a consequence, P line interference is constructive and the R line interferes destructively and the branch may disappear completely[24]. Numerous intensity anomalies occurring due to perturbation have been observed and their intensity patterns

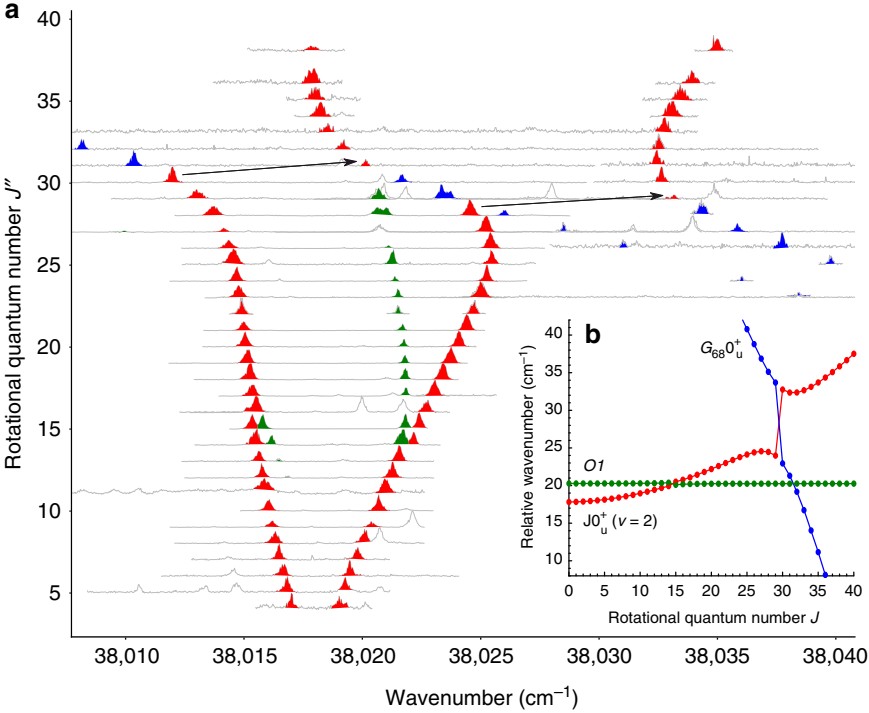

**Fig. 2** An experimental Fortrat diagram. **a** A series of two-colour resonant four-wave mixing (TC-RFWM) spectra ordered by the experimentally determined rotational quantum number $J''$ of the ground state $X0_g^+$ is shown. The regions of unambiguously assigned lines of the $^{65}Cu^{63}Cu$ isotopologue are coloured based on the excited electronic state (blue: $G_{68}0_u^+$, green: $O1$ and red: $J0_u^+$ ($v = 2$)). **b** The corresponding rovibronic levels are plotted, based on molecular constants fitted to the assigned line positions. For clarity, the energy scale was reduced by taking into account the rotational constant $B$ of one state to straighten the parabolas

contain information on the class of perturbation (ref. [24] and references therein). Additional information on perturbation effects is accessible by considering saturation and polarisation features of the background-free double-resonant method applied in this work. In a simplified spectrum, exhibiting only few transitions owing to the stringent double-resonance selection rules, partially forbidden features (weak "extra lines" obtaining oscillator strength through perturbation) can be observed at high laser intensities. Even though bright transitions might substantially broaden upon saturation, they are in general well separated from the "extra lines" that are made visible at increased laser powers. Furthermore, specific linear and circular polarisation configurations of the two resonant lasers allow further enhancement or discrimination of entire families of rovibronic transitions[25] for the characterisation of perturbation effects. Considering the quantum mechanical interference for perturbed levels involving perpendicular transitions, $L$-uncoupling is suggested for the $O1 \sim G_{68}0_u^+$ system and consequently a $1_u$ symmetry label for the $O1$ state. However, the classification of these perturbations requires more detailed experiments and is beyond the scope of this report.

**Isotope-selective tracing of dark bands.** As the dark states exhibit different properties than the bright states, the resulting mixed states can also act as gateways for perturbation-facilitated studies of other dark regions of the studied system[26]. For the copper dimer, a good example is provided by the series of vibrational levels that were preliminarily assigned to the $G0_u^+$ state. In a previous work, we could already affirm computationally that the $B0_u^+$ state and the $G0_u^+$ state emerge from the avoided crossing between a covalent bound state and an ion-pair state[14]. Transitions into ion-pair states are known for their high

transition strength. As a consequence, strong emission into the vibronic ground state $X0_g^+$ ($v = 0$) is observed from the $B0_u^+$ state, which has ion-pair character at the inner potential wall (towards low internuclear distance). By implication, the $G0_u^+$ state should provide similar properties at its outer potential wall. Owing to the shallow potential, this wall is located at a large internuclear distance, so that no vertical transitions into $X0_g^+$ ($v = 0$) are possible. On the contrary, strong transitions from the outer potential wall of $G0_u^+$ would, if they exist, occur into highly excited vibrational levels of $X0_g^+$. While scanning the (0-0) $J0_u^+$-$X0_g^+$ band in the deep-ultraviolet range, we indeed found strong blue fluorescence, which corresponds to transitions into the ground state $X0_g^+$ at vibrational levels starting at $v = 83$. Further analysis revealed that the vibronic state $G_{62}0_u^+$, which is shown in red in Fig. [1], is responsible for this. Only lines belonging to or perturbed by $G_{62}0_u^+$ exhibit this fluorescence. Figure [3] depicts excitation scans in which a spectrometer was used to split the laser-induced fluorescence into individual vibronic bands. In Fig. [3]a, the ultraviolet emission close to the excitation wavelength was chosen. Most extra features visible in Fig. [1]a disappear and just some particular changes in intensity indicate that this still is a heavily perturbed band. Taking a look at the blue fluorescence around $\approx 452$ and 446 nm in Fig. [3]b, c, respectively, it becomes obvious why these features were missing. Already $J0_u^+$ levels that are sharing only a fraction of $G_{62}0_u^+$ character strongly emit in the blue and deplete the excited population rapidly. Therefore, the weaker UV fluorescence is strongly suppressed.

The fast emission of a photon in a totally different spectral range gives these excitation scans a high sensitivity for detecting the dark states. While the nonlinear four-wave mixing discloses the strongly perturbed lines of the dark states close to their intersection with the bright state, the linear, incoherent method is

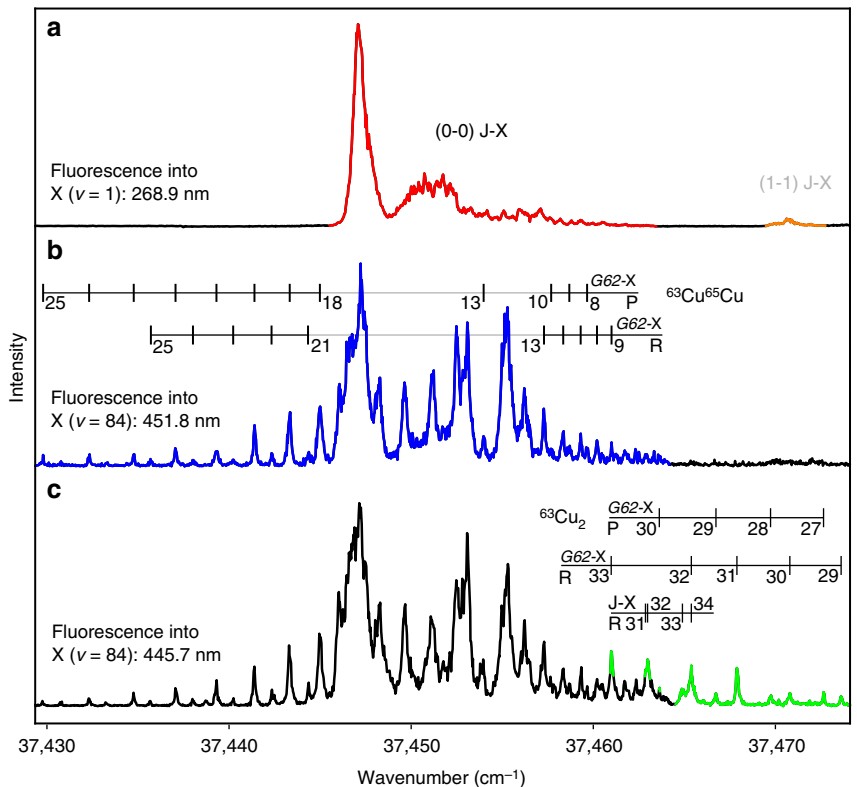

**Fig. 3** Spectral decomposition of the laser-induced fluorescence. Excitation scans around the (0-0) $J0_u^+$-$X0_g^+$ region of the copper dimer. **a** Fluorescence of $J0_u^+$, $v = 0$ into the first vibrational excited level $v = 1$ of the ground state $X0_g^+$ (red). In addition, the (1-1) hot band is weakly visible, as its emission into $v = 2$ occurs just 0.2 nm below (orange). **b** Fluorescence into $v = 88$. Only the $^{65}Cu^{63}Cu$ isotopologue is visible (blue). **c** Fluorescence into $v = 84$. Both isotopologues, $^{65}Cu^{63}Cu$ (black) and $^{63}Cu_2$ (green), are observed simultaneously. Assignments indicate the rotational quantum numbers of the $J0_u^+$ and $G_{62}0_u^+$ to $X0_g^+$ transitions for two isotopologues, which are defined by taking into account the unambiguous results obtained by four-wave mixing

more suitable for the observation of the weak transitions that are energetically more separated. TC-RFWM intensities depend quadratically on the population whereas laser-induced fluorescence intensities depend linearly, which is advantageous in the low-density environment of a molecular beam. In addition, the nonlinear method depends on different powers of the transition moment as the level of saturation changes[27] and is therefore further limiting the detection of weak transitions.

In Fig. 3b, c numerous rotational levels of the dark $G_{62}0_u^+$ state are observed that are below the detection limit of four-wave mixing. The excitation scan in Fig. 3b is obtained by monitoring the emission into the highly excited $v = 88$ level of the ground state $X0_g^+$ in the $^{65}Cu^{63}Cu$ isotopologue. The scan displays rotational levels of the $G_{62}0_u^+$ dark state up to $J' = 25$. The isotope shift of $v = 84$ in the ground state is not sufficiently large to separate the emission of the two main isotopologues by the limited resolution of the spectrometer. As a consequence, weak $^{63}Cu_2$ rotational P and R branch transitions up to $J' = 33$ of $G_{62}0_u^+$ are observed in addition as shown in the Fig. 3c. In spite of the complex appearance of the excitation scans, the assignment of the dark states transitions is straightforward on the basis of the more intense transitions to the dark state that are unambiguously defined by double-resonance labelling.

**Visualisation and deperturbation**. By combining TC-RFWM and excitation scans, we could unambiguously assign approximately 1600 rovibronic lines in the vicinity of the $J0_u^+$ state of $^{63}Cu_2$ and $^{65}Cu^{63}Cu$. While the complete line list is appended as Supplementary Data 1, Fig. 4 illustrates the data graphically by

compiling all scans into a single plot. Only non-overlapping line positions were assigned, though some lines appear to overlap in this depiction. Many of the reported lines were assigned more than once. While this improves the statistics in determination of the line position, the invariance of a line on change of the double-resonant excitation scheme also guarantees the correct assignment. As different excitation schemes involve different overlaps in the transition used for selecting the rotational quantum state, only the correctly assigned lines will remain at the same position.

Next to transitions into the three lowest vibrational bands of the electronic $J0_u^+$ state (shown in shades of blue), Fig. 4 is dominated by transitions into two vibrational levels of the $I1_u$ state (shown in crimson tones). As the vibrational ground state of the $I1_u$ state was never observed, we use preliminary labels for its levels. $I_11_u$ and $I_21_u$ correspond to $v = "x + 1"$ and $v = "x + 2"$, respectively, in the nomenclature used by Powers et al.[19]. While transitions into the $J0_u^+$ state, and all dark states borrowing intensity from it, only exhibit P and R branches, the $I1_u$ state, whose different symmetry we already described in previous work[18], also exhibits a Q branch. Brought to light by the excitation scans, the $G_{62}0_u^+$ and $G_{65}0_u^+$ bands fill significant image area, mostly by their raw spectra plotted in grey. The other $G0_u^+$ bands, as well as the unrelated $L1$, $M1$, $N1$ and $O1$ bands, were only measured close to their intersection with the electronic $J0_u^+$ state. The interaction of the $G0_u^+$ levels with the electronic $J0_u^+$ state increases for higher vibrational excitation, therefore, at higher vibrational levels $v$ of the $J0_u^+$ state, an increasing number of rotational levels became visible also when employing four-wave mixing spectroscopy. In case of $G_{67}0_u^+$ some levels became visible even without an actual intersection with the $J0_u^+$ ($v = 2$) state.

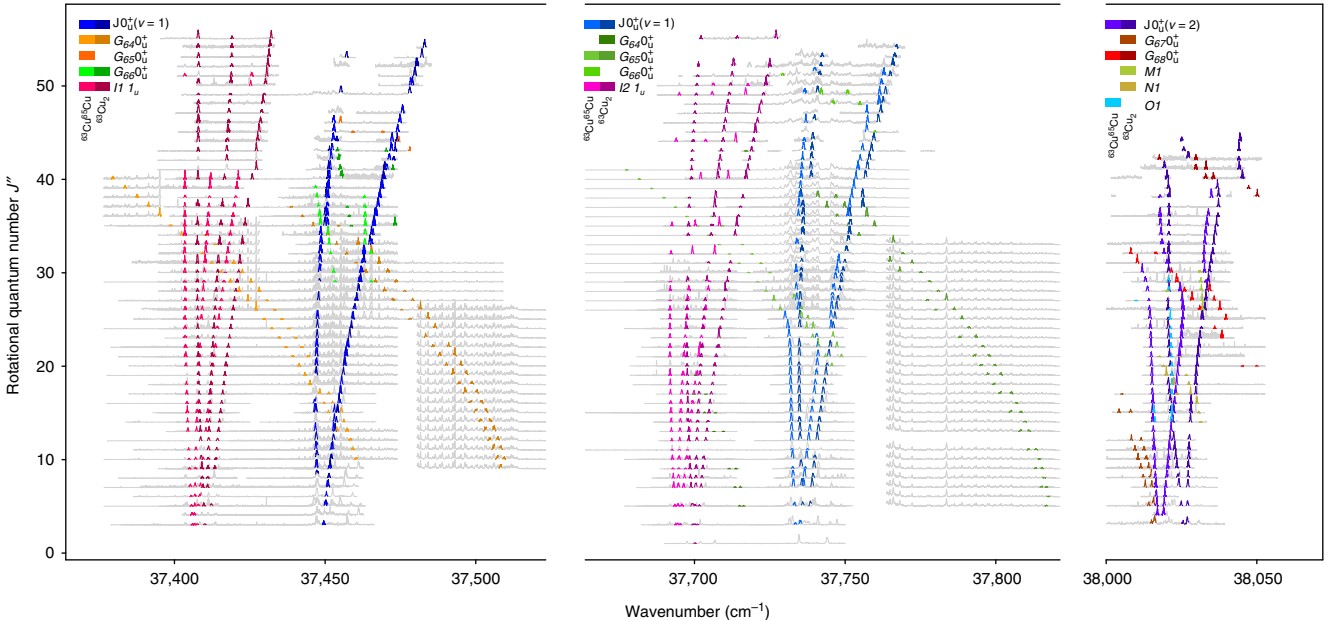

**Fig. 4** Overview of all assigned lines. Total view of the transitions that were rotationally assigned in this work. The regions containing assigned lines are printed colour-coded, while the containing raw spectra are printed in grey. The colours denote the isotopologues and the vibronic states that were accessed by transitions from the vibronic ground state $X0_g^+$ ($v = 0$) with the corresponding rotational quantum number $J''$ as denoted on the ordinate. Starting from the bright transitions that were unambiguously assigned by double-resonant selection of $J''$, blocks of excitation scans into the dark regimes could be also assigned by numbering consecutively

This was additionally favoured by a high thermal population in the observed rotational range. Sufficient $J'$-levels were obtained in this system to evaluate the symmetry of the perturber (see methods section). Based on the experimentally obtained line positions, we fitted molecular constants for the involved states. For the $J0_u^+$ state and the perturbing dark states, a deperturbation was possible. By this, we were able to not only obtain the molecular constants of these states, but also their interaction strengths, as stated in Supplementary Table 1. For the $G0_u^+$ and $L$ levels, also the calculated isotope shifts could be utilised for verification. For the $J0_u^+$ state, the data of both isotopologues were combined to refine the equilibrium constants. The resulting equilibrium constants for $^{63}Cu_2$ are presented in Supplementary Table 2. More details are provided in the methods section. The $I1_u$ state is an anomaly in many respects. Only a few vibrational levels have been observed and even these are too weak to be observed by most methods. We previously reported its short fluorescence lifetime and molecular constants of seemingly unperturbed regions[18]. The $I1_u$ state seems not to interact with the other states characterised in this work, but nevertheless is subject to a complicated pattern of perturbations. Therefore, the $I1_u$ state should be revisited in a subsequent study, so that the perturbing states may be investigated more thoroughly. Such a study can also help to assign the unrelated bands ($L1$, $M1$, $N1$ and $O1$) to specific electronic states. To be used for experimental purposes only, molecular constants that reproduce the observed range were added as Supplementary Table 3.

## Discussion

TC-RFWM spectroscopy allows overcoming the hindrances that have so far prevented experimental access to systems exhibiting a high density of states. Perturbations identified by this method can be used as gateways for experimental access into otherwise inaccessible regions, including the near-dissociation regime of the ground state. These mixed states oscillate between high- and low-vibrational quanta for large and small internuclear distances,

respectively. While at short distance, a model based on the linear combination of atomic orbitals is appropriate, ligand-field theory might be better suited to address the behaviour at long range[28]. The large size of the 4s and 4p orbitals serves to de-shield the atomic-ion cores from each other, leading to a positive point charge perturbing the 3d structures on the other atom. This relates to the foundational concept of "oxidation state", which could be more properly thought of as the charge on one atom as seen by the atom to which it is bound (Field, R. W. (personal communication (2019))). The features of the nonlinear method open a wide field for future experiments on transition metal systems. The presented data also provide reference data to test new quantum chemical methods for computational handling of systems with a high density of states. Such insights lead to fundamental understanding of the complex electronic structure of transition metal clusters that govern their functionality. Ion-pair states, respectively, the states that originate from their avoided crossings with other states, affect almost all regions of the potential energy landscape of the copper dimer. Considering the commonness of perturbation-facilitated mixing between these and other quantum states, it is unlikely that other transition metal systems, which usually exhibit an even higher density of states, can be understood based on the assumption of pure states and the usage of the Born-Oppenheimer approximation. Even for the copper dimer itself, a glance behind the veil of overlapping lines exposed not only a complex network of perturbations, but also vibronic bands displaying interesting properties to access regions of the energy map beyond the possibilities of more conventional spectroscopic methods.

## Methods

**Set-up, spectroscopy and deperturbation.** The copper dimers are generated in a home-built laser-ablation source[18]. Upon ablation with a pulsed Nd:YAG laser (532 nm, 10 ns, 100 mJ per pulse), the plume of copper plasma is entrained in a helium pulse and expanded into high vacuum. In the course of this process condensation and cooling takes place to create a collision-free molecular beam containing sufficient $Cu_2$ for nonlinear spectroscopy. TC-RFWM is performed 1 cm downstream from the nozzle by focusing three parallel propagating laser beams by

a lens ($f = 1000$ mm) into the interaction region. Doppler broadening is eliminated by probing the molecular beam orthogonally to the direction of propagation. If the OODR selection rules[22] are met, a signal is produced in forward direction governed by phase-matching conditions and detected essentially background-free by a photomultiplier tube. The incident laser beams are produced by using Nd:YAG pumped dye lasers with a specified line width of 0.04 cm$^{-1}$. For I1$_u$-X0$_g^+$ and J0$_u^+$-X0$_g^+$ transitions the output of the dye-laser frequency is doubled by second harmonic generation and split to provide the two pump beams of $\approx 100$ μJ per pulse. The (1-0) B-X transitions exhibit a high transition strength and are addressed by the probe beam with laser energies of <100 nJ per pulse. An additional channel of information is obtained by placing a spherical mirror perpendicularly to both molecular beams and the main axis of spectroscopy. The emitted light is focused onto the slit of a 1 m spectrometer in front of a photomultiplier tube.

The assignment of line positions within the raw spectra, as well as the fitting of molecular constants, were performed using the PGOPHER[29] software. Shifted transitions and extra lines caused by perturbations are often revealed by TC-RFWM in a straightforward manner. The stringent double-resonance selection rules yield simplified spectra and allow unambiguous assignments[26,30–33]. The quantification of the perturbation was also based on PGOPHER, where, once the constants were fitted based on the experimental spectra, perturbations can be excluded for comparison. The molecular constants of the ground state of the copper dimer were taken from Ram et al.[34]

**Equilibrium constants of the J0$_u^+$ state.** Utilising the well-established relationships between the molecular constants of different isotopologues[35], we merged the deperturbed constants of both isotopologues before fitting equilibrium constants. While for most constants, the knowledge of the isotopes' masses is sufficient, the determination of the electronic term $T_e$ depends on the knowledge of the ground state zero-point energy, as all experimental data is referenced against it. Based on the Dunham coefficients reported by Ram et al.[34], we calculated the zero-point contributions for $^{63}$Cu$_2$ (132.9711(19) cm$^{-1}$) and $^{65}$Cu$^{63}$Cu (131.9360(18) cm$^{-1}$). After merging the isotopologues, $T_e$, $\omega_e$ and $\omega_e x_e$ were fitted to the $T_v$ values and $B_e$, and $\alpha_e$ were fitted to the $B_v$ values. The equilibrium value for the internuclear distance $r_e$ was then calculated based on $B_e$ and the isotopic masses. To stay consistent within literature, Supplementary Table 2 presents the equilibrium constants in terms of the lightest isotopologue $^{63}$Cu$_2$. However, by using the same relationships, the constants for any isotopologue can be derived. For $^{65}$Cu$^{63}$Cu, for example, one obtains slightly lower values for $\omega_e$ (287.67(17) cm$^{-1}$), $\omega_e x_e$ (1.054 (57) cm$^{-1}$), $B_e$ (0.114276(11) cm$^{-1}$) and $\alpha_e$ (0.0008913(75) cm$^{-1}$).

**Deperturbation and classification of the G$_{67}$0$_u^+$ state.** The G$_{xx}$0$_u^+$ ∼ J0$_u^+$ perturbation strength listed in Supplementary Table 1 were obtained by assuming a homogeneous, $J$ independent perturbation. Trials performed using a heterogeneous perturbation yielded significantly larger residuals for the G$_{67}$0$_u^+$ ∼ J0$_u^+$ ($v=2$) system. A graphical perturbation analysis has been proposed by Bender[36,37] and is shown in the Supplementary Fig. 1. The effective rotational constants, $B_{eff}$, of the G$_{67}$0$_u^+$ ∼ J($v=2$) system vs. $J$ are plotted. The crossing of the traces occurs at $J<0$ and corroborates the homogeneous perturbation mechanism. Since the symmetry of the bright state J0$_u^+$ is certainly an $\Omega = 0_u^+$ state, a homogeneous perturbation determines the perturber G$_{67}$0$_u^+$ and the other members of the G$_{xx}$0$_u^+$ progression as 0$_u^+$ states. For comparison, a homogeneous perturbation model is applied for the deperturbation of the L1, M1, N1 and O1 perturber states. A definitive assignment, however, requires more detailed investigations.

**Vibrational quantum numbers of the G$_{xx}$0$_u^+$ states.** The deperturbed molecular constants in Supplementary Table 1 allowed determination of the absolute vibrational numbering. The presumption that the vibrational progression of the G$_{62}$0$_u^+$ to G$_{68}$0$_u^+$ levels belong to the G0$_u^+$ state was supported by a good qualitative match between their spacing and the expected spacing extrapolating from the reported levels from Powers et al.[19] Following the established approach of determining the vibrational level $v$ based on the isotope shift[35], we fitted our values to a Dunham expansion, which led to an absolute value of about $v = 62$ for the lowest of these levels. The initial approach in doing so was to use an equilibrium term energy of $T_e = 30695$ cm$^{-1}$. This value was derived from the levels reported by Powers et al., but taking into account a correction of the numbering assignments ($v_N \rightarrow v_{N-1}$) already proposed by Rohlfing et al.[38] However, defining $T_e$ as a free fit parameter, the obtained value $v = 62$ barely changed (e.g., 62.28 → 62.33). This is in good agreement with recent work by Pashov et al.[39], in which they demonstrate the establishment of absolute vibrational numbering based on four term energies in one isotopologue and one term energy in another. Also when using their method, which was applied after simulating unperturbed bands using PGOPHER, G$_{62}$0$_u^+$ was also assigned to $v = 62$. Nevertheless, one should take these values with caution. We have already demonstrated that the low-lying electronic states of the copper dimer can be only explained by assuming spin–orbit states given by Hund's case (c) coupling[14]. In the computational part of that work, only the potential curves of states corresponding to separated atom limits with only a single excitation were fully covered. Therefore, only two covalently bound 0$_u^+$ states interact with the 0$_u^+$ ion-pair ground state to bring forth the A, the B, and the G0$_u^+$ states. Taking a look at the experimentally derived potential energy curves of Cu$_2$, which were

illustrated, e.g., in a book chapter by Morse[15] (p. 95), it becomes obvious that the separated atom limits with double excitation cannot be ignored, and further 0$_u^+$ states originate from the $^2$D + $^2$D atomic limits. As these states also contribute avoided crossings and, in addition, $v = 0$ of the G0$_u^+$ state was never observed, it is still too early to decide if the methods used above are applicable for this system.

**Interpolation of missing vibrational origins.** Within Supplementary Table 1, only the numbers with a specified standard deviation are directly derived from experimental data. The missing $T_v$ values of G$_{xx}$0$_u^+$ states were filled based on the fitted Dunham expansion that was also used to determine the absolute vibrational levels. For the perturbations, the strengths were similar whenever we had experimental access to both isotopologues. Therefore, we presumed a similarity also when only one isotopologue was measured.

**Franck-Condon overlaps in the G$_{xx}$0$_u^+$ ∼ J0$_u^+$ systems.** Supplementary Fig. 2 shows the RKR potentials of the J0$_u^+$ and G states, which were constructed by performing Dunham fits to the molecular constants listed in the Supplementary Table 1. The wave functions and the overlap integrals are then obtained by solving the radial Schrödinger equation. A ratio of 1:3:6 is computed for the overlaps between G$_{62}$0$_u^+$ ∼ J($v=0$) ≈ G$_{63}$0$_u^+$ ∼ J($v=0$): G$_{64}$0$_u^+$ ∼ J($v=1$) ≈ G$_{65}$0$_u^+$ ∼ J($v=1$) ≈ G$_{66}$0$_u^+$ ∼ J($v=1$): G$_{67}$0$_u^+$ ∼ J($v=2$) ≈ G$_{68}$0$_u^+$ ∼ J($v=2$), which is in good agreement with the values for the homogeneous perturbation strength between the G$_{xx}$0$_u^+$ and J0$_u^+$ states deduced from the experiment and shown in Supplementary Table 1.

**Simulation of perturbed bands.** To generate Fig. 1, we used PGOPHER to simulate the (0-0) J0$_u^+$-X0$_g^+$ band of Cu$_2$ based on the constants and interactions reported in Supplementary Table 1. For the spectrum (Fig. 1a), a rotational temperature of 50 K was chosen, in order to achieve a close resemblance of the intensity distribution observed in our experiment. The abundance of the isotopologues was based on natural abundance. As the different statistical weight of symmetrical and asymmetrical rotational levels (3 and 5) in the homonuclear isotopologue is taken into account in the simulation, an average statistical weight of 4 was chosen for all levels of the heteronuclear isotopologue to match the average intensity. In the Fortrat diagram, straight lines between the points of each branch were added as a guide to the eye. To emphasise the perturbations that yield some obvious spectral features, the corresponding regions of the Fortrat diagram were copied into the simulated spectra.

**Generation of Fig. 2.** Parsing the raw list of all assigned lines, the corresponding raw spectra were loaded and normalised. They were then added to the scalable vector graphics (SVG) file, using the rotational quantum number in the ground state as a vertical offset. Therefore, the peak heights do not scale with the absolute intensity, when comparing different scans. For some rotational levels, different lines were used to select the rotational quantum state. Owing to the different overlaps associated with these lines, only the peaks that belong to the targeted rotational state reproduce. In a second round, windows around the assigned line positions were selected from each raw spectrum. These were redrawn with a colourised background to highlight the assigned peak. To achieve a clearer presentation, a few peaks were manually rescaled and the colorized ranges were slightly readjusted when arranging and formatting the final Fig. 2a. For the corresponding level diagram in Fig. 2b, we used the simulation based on the fitted molecular constants. At the scale employed, this does not affect the position of the data points, but improves clarity by extrapolating points missing within the experimental dataset. To further clarify the level repulsion, the levels were plotted as relative wavenumber $\tilde{\nu}_{relative} = \tilde{\nu}_{absolute} - 38,000\ \text{cm}^{-1} - B_{O1}J(J+1)$ where $B_{O1}$ is the rotational constant of the O1 state as reported in Supplementary Table 1. Therefore, the relative values are referenced against an unperturbed, harmonic version of O1.

**Details on the excitation scans.** When recording the spectrum in Fig. 3a, some stray light of the UV pump laser passed the spectrometer off-axis. Therefore, we started integration 20 ns after the maximum of the pump pulse to ensure that also the electrical response of the photomultiplier had concluded. In contrast, the strong, and therefore short-lived, blue fluorescence shown in Fig. 3b, c was integrated over the complete pulse. This timing difference is responsible for the complete disappearance of the G$_{62}$0$_u^+$-related lines in Fig. 3a.

**Generation of Fig. 4.** This overview figure was created by reusing the code used for Fig. 2, but applying it on the full line list. As this inevitably produces overlapping spectra, coloured lines (instead of coloured fills) were used to indicate the assigned lines. While the full spectral raw data was exposed to the Matplotlib library, some data points that did not contribute to significant changes in the shape of the curve were dropped. Therefore, zooming into the vector graphic will not always show the full details of the background noise. The total absence of background in some of the excitation scans allowed to still obtain clear peaks even when averaging at event rates of less than one event per laser pulse. Because of the normalisation of the plotted raw spectra, these features can be invisible, if there are strong lines in the same spectrum. An extreme example of this are the high rotational levels of the G$_{65}$0$_u^+$ state in the

$^{65}$Cu$^{63}$Cu isotopologue. There, the background is plotted as a completely flat line and the coloured peaks are just visible, because the amplitude was manually increased when composing the final depiction.

## Data availability

The raw spectroscopic data that supports the findings of this study and the resulting PGOPHER files are available from the corresponding author upon reasonable request. The authors declare that all other data supporting the findings of this study are available within the paper and its Supplementary Information files.

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

## Acknowledgements
This work is supported by Swiss National Science Foundation (#200021_1531701 and #200020_1754901).

## Author contributions
M.B., B.V. and P.R. carried out the experiments. P.B., M.B. and P.R. analysed the data. M.B., P.R., P.B. and G.K. interpreted the results. P.R. and J.v.B. supervised the study. M.B. wrote the initial manuscript. All authors discussed and reviewed the manuscript.

## Additional information

**Competing interests:** The authors declare no competing interests.

