## [Peer Review File · Nature Communications]

Review of “Spectroscopic disentanglement of the quantum states of highly excited Cu_2 ,” by Martin Beck, Peter Bornhauser, Bradley Visser, Gregor Knopp, Jeroen A. van Bokhoven, and Peter P. Radi

This manuscript presents a definitive study of the highly excited electronic states of Cu_2 that lie in the $37,400 - 38,050 \text{ cm}^{-1}$ range, using a sophisticated two-color four-wave mixing technique to disentangle the complicated electronic structure in this region. The paper represents very high quality spectroscopic work and is certainly worthy of publication. I have a few suggestions that I think will greatly improve the quality of the manuscript, however.

More important suggestions:

1. On page 3, the authors attempt to make a connection between their work on the dicopper molecule and enzymes that employ dicopper centers. This is really quite a stretch, since the enzymes employ copper in the +1 or +2 oxidation state, and the two copper centers are not chemically bonded to each other directly, but are instead bridged by oxo or hydroxo bridges. In my opinion, it is disingenuous to suggest that anything learned about the Cu_2 molecule will be relevant to dicopper enzymes. I recommend that this section of text be simply deleted.
2. On page 5, in the section “Rotational disentanglement of bands”, the authors first introduce the technique of two-colour resonant four-wave mixing. This has been the key to the success of the present investigation, and in my opinion its virtues could be more clearly identified. I would suggest introducing this section with a statement like “Two-colour resonant four-wave mixing is a double resonance technique that allows spectra to be collected that originate from a single known J'' level of the ground $X^1\Sigma_g^+$ state. The resulting spectra contain only one R, one P, and sometimes one Q line, depending on the symmetry of the excited state. This capability greatly simplifies the analysis of the spectra, especially in cases where perturbations destroy the regularity of rotational levels in the upper state. When a bright state ...”
2. On page 7, the authors state “In contrast, the vibronic state preliminarily labelled O1 exhibits a combination of properties that lies beyond the known effects in diatomic molecules²².” The citation is to the definitive text on diatomic molecule spectroscopy by Lefebvre-Brion and Field. In my opinion, this statement is far too strong. It suggests that there are additional interactions present in diatomic molecules like Cu_2 that have not yet been described or identified in this definitive text. I very much doubt that this is the case. More likely, there are other interactions or electronic states that the authors have not yet recognized, but which when identified will be able to explain all of the experimental observations using the ideas and interactions that are explicated in this definitive text. In my opinion, this statement must be softened. It would be more appropriate to simply say that at this time the authors do not have an explanation for the anomalous properties of the O1 state.
3. I would like to compliment the authors on the very nice description on page 8 of why the G62 level (and other high vibrational levels of the G state) fluoresce rapidly to the X

state with a transition moment that is dominated by the outer limb of the potential surface, while the transition moment for the B-X transition is dominated by a large transition moment on the inner limb of the potential curves. In view of the ion-pair nature of the G state (at large internuclear separations) and the ion-pair nature of the B state (at small internuclear separations), this behavior makes perfect sense. The scans displayed in Figure 3 display the described behavior in a most elegant way.

4. Throughout this paper, the authors neglect to identify the values of the good quantum numbers of the states: Ω , g/u, and for $\Omega=0$, +/- . In order to properly understand these states, I recommend that the authors attach these Ω -labels to the state labels. For example, the interacting J and G states are apparently $\Omega = 0_u^+$ in symmetry. It would be good to label these states as the $J0_u^+$ and $G0_u^+$ states throughout the manuscript. Similarly, the I1 and I2 states are of $\Omega=1_u$ symmetry, and this should be designated as well. Perhaps as $I1_u-1$ and $I1_u-2$ (or something similar). Along these lines, the fitted perturbation matrix elements listed in Table 1 between the vibrational levels of the J state and the L1, M1, N1, and O1 states are treated as constants independent of the rotational quantum number J. This implies that the perturbation coupling the states is homogeneous, and that the L1, M1, N1, and O1 states share the same Ω value as the J state, $\Omega=0_u^+$. If this is correct, the authors should state this explicitly since this narrows down the possible candidates for these states. These labels should be included in all references in the text, in the tables, and in the figures.

5. On page 11, the authors state, "The I state seems not to interact with the other states characterised in this work, but nevertheless is subject to a complicated pattern of perturbations." Given the fact that the I state is of 1_u symmetry and (see comment #4) all of the other states appear to be of 0_u^+ symmetry, the only interactions between these I state and these other states possible would be heterogeneous perturbations that have matrix elements that are roughly proportional to the rotational constant times J, times another factor (which could be significantly smaller than 1). Thus, it may not be surprising that the I state fails to interact with the states that have been analyzed in this paper. It may provide entry into the states of $\Omega = 1_u$ symmetry, however.

The next sentence "Therefore, the I state needs to be revisited by a subsequent study, which preponds population transfer to an intermediate state to then uncover also states that are inaccessible from the ground state." is in my opinion very convoluted and difficult to follow. In part, this is due to the use of the unusual word "preponds". I would suggest revising this sentence to read something like: "Therefore, the I state should be revisited in a subsequent study, so that the perturbing states may be investigated more thoroughly."

6. On page 15, in the section on the Simulation of perturbed bands, the last sentence says "It should be noted that the effect estimated by the $\langle L1||G62 \rangle$ perturbation in Extended Data Table 1 was omitted in this simulation, as the effect of this simplification would contradict the experimental spectrum in Figure 3b." I find it impossible to understand the meaning of this sentence. If inclusion of the effects of this perturbation contradicts the

experimental spectrum, wouldn't that imply that the perturbation is not properly treated? This sentence should be revised to be more clear.

7. I found the details about the generation of the various figures to be overly complicated and detailed. I don't know if the journal requires this level of detail, but I found it to detract from the paper.

Less important corrections:

1. On page 2, change "via catalysis and sensors" to "catalysis and sensors". Later on this same page the authors use the word "manifoldness" in the sentence: "In this regime, the partially filled d orbitals of transition metals are important, as the implied configurational manifoldness gives rise to a large number of electronic states." I don't believe that "manifoldness" is actually a word, so I would recommend that this sentence be rephrased along the lines of: "In this regime, the partially filled d orbitals of transition metals lead to a very large number of electronic states." In the next sentence the authors mention perturbation-induced mixing of energetically close states and in the following sentence mention vibronic coupling but neglect other important perturbations that can cause state mixing. Here I would recommend replacing the sentence with one that includes spin-orbit interaction, reading something like "Such vibronic coupling, which is neglected in the Born-Oppenheimer approximation, along with spin-orbit interaction combines with coherence phenomena to enhance function in chemical and biophysical..."

2. On page 12, 6 lines above the beginning of the Methods section, "others" should be changed to "other", so the sentence reads "...mixing between these and other quantum states..."

Overall Recommendation:

This is a superb spectroscopic study of a highly perturbed section of the spectrum of Cu_2 that has completely eluded analysis up until this point. It should certainly be published after the issues raised above have been addressed. Whether it is of sufficient general interest to merit publication in Nature Communications or whether it should be published in a more specialized journal, such as ChemPhysPhysChem, Molecular Physics, the Journal of Chemical Physics, J. Phys. Chem A, or someplace else that is more of an open question. In any case, the manuscript represents a huge amount of work and the authors are to be commended for cracking the very tough nut that is Cu_2 in this energy range.

Reviewer #2 (Remarks to the Author):

The electronic spectra of all homonuclear transition metal dimers are extremely complicated owing to the very high density of electronic states. The choice of Cu₂ is an inspired one, because the electronic ground state is necessarily simple, thus providing a solid foundation for examination of the horrendous complexity that begins as soon as the 3d shell is opened. This perhaps justifies Nature Communications status for this paper. However, I would have liked to see a more aggressive framework, for both the "big picture" of the electronic structure of transition metal dimers and the methods of assignment.

I suggest adding a paragraph about a physical picture for the electronic structure of Cu₂ and a second paragraph about the important themes in the electronic structure of all transition metal dimers. The 4s orbital has a much larger radius than that of 3d. Forming a bonding molecular orbital out of 4s atomic orbitals results in a very weak bond owing to overlap repulsion of the σ_{4s} orbital with the filled 3d¹⁰ subshell. As a result, Franck-Condon overlap is exclusively into high-vibrational levels of all of the nominal 3d⁹4s² or 3d⁹4s⁴p molecular states. At large internuclear distance, the "big-picture" electronic structure will be described by a form of ligand field theory. The large size of the 4s and 4p orbitals serves to de-shield the atomic-ion cores from each other, leading to a positive point charge perturbing the 3d structures on the other atom. This relates to the foundational concept of "oxidation state," which could be more properly thought of as the charge on one atom as seen by the atom to which it is bound. The 3d⁹4s²-3d¹⁰4s and 3d⁹4s²-3d⁹4s² separated atom limits are very low-lying (~1.5 eV), with respective total degeneracies of 10x2x2 and 102! This gives an enormous number of low-lying electronic states, some of which are much more strongly bound than the ground state (especially for states in which the overlap repulsion is minimized via the hole in the 3d shell). Since these excited states are sampled at high-v via electronic transitions from the ground state, over the oscillation range of each high-v level the electronic structure goes from ligand-field-like at large-R to LCAO-MO-like at small-R.

I suggest adding a paragraph about the history of Perturbation Facilitated Optical-Optical Double Resonance (PF-OODR) spectroscopy. PF-OODR spectroscopy is not a new area of research. Early papers on the subject date from the 1970's (Schawlow, Field, many others). Marjatta Lyyra has made a career out of perturbation-facilitated all optical triple resonance (AOTR) schemes. The sensitivity of four-wave mixing (4WM) schemes relative to fluorescence- or ionization-detected incoherent double resonance (OODR) schemes has been discussed by Rohlfing and others. 4WM is background-free so it is more sensitive for strong transitions than the incoherent schemes, but the incoherent schemes are more sensitive for weak transitions because the signal dependences on different powers of transition moments, which is of extreme importance for observation of weak extra lines at perturbations.

I suggest adding a paragraph about Spectroscopic diagnostics for multiple resonance schemes and for classes of perturbations. For example, polarization effects affect the intensity ratio for PR,PR and Q,Q transition sequences with parallel polarizations of the two lasers and PR,Q for perpendicular polarizations. Owing to the opposite signs of R vs. P transition amplitudes for perpendicular type ($\Delta J = \pm 1$) transitions vs. same signs for parallel ($\Delta J = 0$) transitions, there will be anomalies in the R/P intensity ratio when there is an L-uncoupling ($\Delta J = \pm 1$) [B.J.L] perturbation. These perturbations have matrix elements that are approximately linearly dependent on J, thus the size and even the sign of the R/P intensity anomaly becomes informatively J-dependent. There is a brief remark in the present text raising the possibility of such effects, but without any logical or useful structure.

What about nuclear permutation symmetry ortho:para intensity ratios for the 63-63 isotopologue and none for the 63-65? What about some discussion of what makes a state dark or bright with respect to a particular excitation path? What about discussion of the calculable factors (vibrational overlap integral, interrelationships between spin-orbit and L-uncoupling matrix elements) that go into the value of a particular perturbation matrix element?

Specific Details:

1. Page 2. Manifoldness is not an English word.
2. Figure 1 caption and page 5. Define Fortrat parabola.
3. Page 5. Detectable (not detectible).
4. Page 6. The state name G68 appears with little explanation.
5. Page 6. "...swap their energy order..."
6. Page 7. "...beyond the known effects in diatomic molecules..." is a ridiculous and untrue statement, certainly not supported by reference 22.
7. Page 7. "...expected for a dark state..."
8. Page 7. Some needlessly vague and all-encapsulating statements about "intensity anomaly", "unexplained interference effects," and "polarization dependent studies".
9. Page 7. "...preliminarily..."
10. Page 8. "Taking a look at the blue..." Blue in the spectrum or in the figure?
11. Page 9. "...increasing the number of samples averaged at each data point" What does this mean?
12. Page 9. the discussion of isotope shifts is clumsy and needlessly opaque.

13. Page 10-11. "...increases for higher vibrational excitation..." Is this a vibrational overlap or a J-dependent matrix element effect?
14. Page 11. The nature of the "anomaly" exhibited by the I state is unclear. Is the short lifetime due to predissociation or to an unexpectedly strong radiative transition?
15. Page 11. "Prepends" is not an English word.
16. Page 12. I like "a glance behind the veil." But I do not like the double vagueness of "unexpected properties to be resolved by a new generation of methods."

This is a beautiful paper, but it could be much more beautiful if it could be modified to include some of my suggestions. I suggest that, after modifications, it will be well suited for a Nature Communication.

Robert W. Field

Reviewer #3 (Remarks to the Author):

Beck et al. present a spectroscopic study of electronic transitions of the Cu₂ dimer that are observed in the 37350-30850 cm⁻¹ range. They have used a two-dimensional technique to unravel the complex rotational band structures observed at these energies. This work is an extension of their previous 2-dimensional study of the same bands that was published in the Journal of Physical Chemistry A (reference 21). The present work reports additional measurements and a careful deperturbation analysis. While this is certainly a respectable study, the novelty is oversold. Many previous studies have shown the value of two-dimensional techniques for electronic spectroscopy, and the fact that Cu₂ has a high density of states above 30000cm⁻¹ is no surprise. I would not consider the intensity anomalies of transitions to the O1 state to be "beyond the known effects in diatomic molecules." In general, theoretical models of perturbation-induced intensity anomalies are well-developed. In summary, this is a quality spectroscopic study that provides new spectroscopic data for the Cu₂ dimer. I recommend publication after some softening of the claims of novelty.

Reply to the Reviewers

We greatly appreciate the careful review of our manuscript and the highly valued suggestions. We tried to address all issues detailed and carefully as documented in the following. With the considerable effort of the reviewers, the quality of the paper could be improved substantially. The answers to the reproduced comments are shown in blue below.

Reviewer #1

Dear Reviewer,

we would like to thank, in particular, for the statements in issue #3 and the last sentence of the Overall recommendation.

Review of “Spectroscopic disentanglement of the quantum states of highly excited Cu₂,” by Martin Beck, Peter Bornhauser, Bradley Visser, Gregor Knopp, Jeroen A. van Bokhoven, and Peter P. Radi. This manuscript presents a definitive study of the highly excited electronic states of Cu₂ that lie in the 37,400 – 38,050 cm⁻¹ range, using a sophisticated two-color four-wave mixing technique to disentangle the complicated electronic structure in this region. The paper represents very high quality spectroscopic work and is certainly worthy of publication. I have a few suggestions that I think will greatly improve the quality of the manuscript, however.

More important suggestions:

1. On page 3, the authors attempt to make a connection between their work on the dicopper molecule and enzymes that employ dicopper centers. This is really quite a stretch, since the enzymes employ copper in the +1 or +2 oxidation state, and the two copper centers are not chemically bonded to each other directly, but are instead bridged by oxo or hydroxo bridges. In my opinion, it is disingenuous to suggest that anything learned about the Cu₂ molecule will be relevant to dicopper enzymes. I recommend that this section of text be simply deleted.

A: deleted

2. On page 5, in the section “Rotational disentanglement of bands”, the authors first introduce the technique of two-colour resonant four-wave mixing. This has been the key to the success of the present investigation, and in my opinion its virtues could be more clearly identified. I would suggest introducing this section with a statement like “Two-colour resonant four-wave mixing is a double resonance technique that allows spectra to be collected that originate from a single known J level of the ground X^{1g+} state. The resulting spectra contain only one R, one P, and sometimes one Q line, depending on the symmetry of the excited state. This capability greatly simplifies the analysis of the spectra, especially in cases where perturbations destroy the regularity of rotational levels in the upper state. When a bright state ...”

A: We have modified the paragraph according to this suggestion.

2. On page 7, the authors state “In contrast, the vibronic state preliminarily labelled O1 exhibits a combination of properties that lies beyond the known effects in diatomic molecules²².” The citation is to the definitive text on diatomic molecule spectroscopy by Lefebvre-Brion and Field. In my opinion, this statement is far too strong. It suggests that there are additional interactions present in diatomic molecules like Cu₂ that have not yet been described or identified in this definitive text. I very much doubt that this is the case. More likely, there are other interactions or electronic states that the authors have not yet recognized, but which when identified will be able to explain all of the experimental observations using the ideas and interactions that are explicated in this definitive text. In my opinion, this statement must be softened. It would be more appropriate to simply say that at this time the authors do not have an explanation for the anomalous properties of the O1 state.

A: The paragraph has been completely rewritten and softened

The observation of the O1 perturbation in subsection Rotational disentanglement of bands in Results is followed by

Intensity anomalies between $P(J+1)$ and $R(J-1)$ lines have been observed and attributed to the mixing of perturbing states.[1] The effect can be explained by quantum mechanical interference that comprises information on the perturbation class. For example, parallel and perpendicular transitions (with an orbital angular momentum change $\Delta\Lambda = 0$ or ± 1 , respectively) display approximately equal intensities for the $P(J+1)$ and $R(J-1)$ lines for unperturbed states. If a perturbation occurs (L -uncoupling) between two states exhibiting an angular momentum difference of $\Delta\Lambda = \pm 1$, the relative intensity of the branches out of this state can be affected strongly. For parallel transitions the amplitude phases of P and R are identical while for perpendicular transitions they are opposite. As a consequence, P line interference is constructive and the R line interferes destructively and the branch may disappear completely.[2] Numerous intensity anomalies occurring due to perturbation have been observed and their intensity patterns contain information on the class of perturbation (Ref. [2] and references therein). Additional information on perturbation effects is accessible by considering saturation and polarization features of the background-free double-resonant method applied in this work. In a simplified spectrum, exhibiting only few transitions owing to the stringent double-resonance selection rules, partially forbidden features (weak 'extra lines' obtaining oscillator strength through perturbation) can be observed at high laser intensities. Even though bright transitions might substantially broaden upon saturation they are in general well separated from the 'extra lines' that are made visible at increased laser powers. Furthermore, specific linear and circular polarization configurations of the two resonant lasers allow further enhancement or discrimination of entire families of rovibronic transitions[3] for the characterization of perturbation effects. Considering the quantum mechanical interference for perturbed levels involving perpendicular transitions, l-uncoupling is suggested for the $O1 \sim G_{68}0_u^+$ system and consequently a 1_u symmetry label for the $O1$ state. However, the classification of this perturbations requires more detailed experiments and is beyond the scope of this report.

3. I would like to compliment the authors on the very nice description on page 8 of why the G62 level (and other high vibrational levels of the G state) fluoresce rapidly to the X state with a transition moment that is dominated by the outer limb of the potential surface, while the transition moment for the B-X transition is dominated by a large transition moment on the inner limb of the potential curves. In view of the ion-pair nature of the G state (at large internuclear separations) and the ion-pair nature of the B state (at small internuclear separations), this behavior makes perfect sense. The scans displayed in Figure 3 display the described behavior in a most elegant way.

4. Throughout this paper, the authors neglect to identify the values of the good quantum numbers of the states: J , g/u , and for $=0, +/-$. In order to properly understand these states, I recommend that the authors attach these $-$ labels to the state labels. For example, the interacting J and G states are apparently $= 0u+$ in symmetry. It would be good to label these states as the $J0u+$ and $G0u+$ states throughout the manuscript. Similarly, the I1 and I2 states are of $=1u$ symmetry, and this should be designated as well. Perhaps as $I1u-1$ and $I1u-2$ (or something similar). Along these lines, the fitted perturbation matrix elements listed in Table 1 between the vibrational levels of the J state and the L1, M1, N1, and O1 states are treated as constants independent of the rotational quantum number J . This implies that the perturbation coupling the states is homogeneous, and that the L1, M1, N1, and O1 states share the same J value as the J state, $=0u+$. If this is correct, the authors should state this explicitly since this narrows down the possible candidates for these states. These labels should be included in all references in the text, in the tables, and in the figures.

A: We have added the Omega labeling as suggested for G, J and I. For L1, M1, N1 and O1 we applied a homogeneous perturbations for comparison with the determined homogeneous perturbations in the J G system. A sentence is added to Supplementary Data Table I and to the text in Section Deperturbation and classification of the $G_{67}0u+$ state

5. On page 11, the authors state, "The I state seems not to interact with the other states characterised in this work, but nevertheless is subject to a complicated pattern of perturbations." Given the fact that the I state is of $1u$ symmetry and (see comment #4) all of the other states appear to be of $0u+$ symmetry, the only interactions between these I state and these other states possible would be heterogeneous perturbations that have matrix elements that are roughly proportional to the rotational constant times J , times another factor (which could be significantly smaller than 1). Thus, it may not be surprising that the I state fails to interact with the states that have been analyzed in this paper. It

may provide entry into the states of $= 1u$ symmetry, however. The next sentence “Therefore, the I state needs to be revisited by a subsequent study, which prepends population transfer to an intermediate state to then uncover also states that are inaccessible from the ground state.” is in my opinion very convoluted and difficult to follow. In part, this is due to the use of the unusual word “prepends”. I would suggest revising this sentence to read something like: “Therefore, the I state should be revisited in a subsequent study, so that the perturbing states may be investigated more thoroughly.”

A: We appreciate greatly the remark on the possibility of only weak perturbations of the I state. The sentence has been modified as suggested.

6. On page 15, in the section on the Simulation of perturbed bands, the last sentence says “It should be noted that the effect estimated by the 1— 62 perturbation in Extended Data Table 1 was omitted in this simulation, as the effect of this simplification would contradict the experimental spectrum in Figure 3b.” I find it impossible to understand the meaning of this sentence. If inclusion of the effects of this perturbation contradicts the experimental spectrum, wouldn’t that imply that the perturbation is not properly treated? This sentence should be revised to be more clear.

A: There might be an additional perturbation on the perturber state. But the perturbation is negligible in our context. The sentence has been dropped.

7. I found the details about the generation of the various figures to be overly complicated and detailed. I don’t know if the journal requires this level of detail, but I found it to detract from the paper.

A: We will check with the editor to what level the information needs to be provided

Less important corrections:

1. On page 2, change “via catalysis and sensors” to “catalysis and sensors”.

A: changed

Later on this same page the authors use the word “manifoldness” in the sentence: “In this regime, the partially filled d orbitals of transition metals are important, as the implied configurational manifoldness gives rise to a large number of electronic states.” I don’t believe that “manifoldness” is actually a word, so I would recommend that this sentence be rephrased along the lines of:

“In this regime, the partially filled d orbitals of transition metals lead to a very large number of electronic states.”

A: changed

In the next sentence the authors mention perturbation-induced mixing of energetically close states and in the following sentence mention vibronic coupling but neglect other important perturbations that can cause state mixing. Here I would recommend replacing the sentence with one that includes spin-orbit interaction, reading something like

“Such vibronic coupling, which is neglected in the Born-Oppenheimer approximation, along with spin-orbit interaction combines with coherence phenomena to enhance function in chemical and biophysical...”

A: modified

2. On page 12, 6 lines above the beginning of the Methods section, “others” should be changed to “other”, so the sentence reads “...mixing between these and other quantum states...”.

A: modified

Overall Recommendation:

This is a superb spectroscopic study of a highly perturbed section of the spectrum of Cu₂ that has completely eluded analysis up until this point. It should certainly be published after the issues raised above have been addressed. Whether it is of sufficient general interest to merit publication in Nature Communications or whether it should be published in a more specialized journal, such as ChemPhysPhysChem, Molecular Physics, the Journal of Chemical Physics, J. Phys. Chem A, or someplace else that is more of an open question. In any case, the manuscript represents a huge amount of work and the authors are to be commended for cracking the very tough nut that is Cu₂ in this energy range.

Reviewer #2

Dear Prof. Robert Field

Please note that we cited and referenced (Ref[33]) your comment on the foundational concept of 'oxidation state' and hope that you agree with the insertion.

The electronic spectra of all homonuclear transition metal dimers are extremely complicated owing to the very high density of electronic states. The choice of Cu₂ is an inspired one, because the electronic ground state is necessarily simple, thus providing a solid foundation for examination of the horrendous complexity that begins as soon as the 3d shell is opened. This perhaps justifies Nature Communications status for this paper. However, I would have liked to see a more aggressive framework, for both the "big picture" of the electronic structure of transition metal dimers and the methods of assignment.

R1: I suggest adding a paragraph about a physical picture for the electronic structure of Cu₂ and a second paragraph about the important themes in the electronic structure of all transition metal dimers. The 4s orbital has a much larger radius than that of 3d. Forming a bonding molecular orbital out of 4s atomic orbitals results in a very weak bond owing to overlap repulsion of the 4s orbital with the filled 3d¹⁰ subshell. As a result, Franck-Condon overlap is exclusively into high-vibrational levels of all of the nominal 3d⁹4s² or 3d⁹4s⁴p molecular states. At large internuclear distance, the 'big-picture' electronic structure will be described by a form of ligand field theory. The large size of the 4s and 4p orbitals serves to de-shield the atomic-ion cores from each other, leading to a positive point charge perturbing the 3d structures on the other atom. This relates to the foundational concept of 'oxidation state,' which could be more properly thought of as the charge on one atom as seen by the atom to which it is bound. The 3d⁹4s²-3d¹⁰4s and 3d⁹4s²-3d⁹4s² separated atom limits are very low-lying (1.5 eV), with respective total degeneracies of 10x2x2 and 102! This gives an enormous number of low-lying electronic states, some of which are much more strongly bound than the ground state (especially for states in which the overlap repulsion is minimized via the hole in the 3d shell). Since these excited states are sampled at high-v via electronic transitions from the ground state, over the oscillation range of each high-v level the electronic structure goes from ligand-field-like at large-R to LCAO-MO-like at small-R.

A1: We have modified large parts of the manuscript and tried to provide a more 'aggressive' picture of the electronic structure of Cu₂.

The Introduction is completely rewritten and reflect in more detail the electronic structure and challenges of TM in general and of Cu₂ in particular. Unfortunately, a more extended overview is not possible due to the 1000-word restriction of the journal. We have included the possibility of addressing the electronic structure at large internuclear distances by applying ligand-field theory in the Discussion (formerly: Conclusions) section. The J~G perturbation is a very nice example of how to access regions of the energy map that are not accessible by more conventional methods.

R2: I suggest adding a paragraph about the history of Perturbation Facilitated Optical-Optical Double Resonance (PF-OODR) spectroscopy. PF-OODR spectroscopy is not a new area of research. Early papers on the subject date from the 1970's (Schawlow, Field, many others). Marjatta Lyyra has made a career out of perturbation-facilitated all optical triple resonance (AOTR) schemes.

A2: We did not apply PFOODR in this work and a paragraph on the topic would confuse readers. However, in a subsequent paper, we will focus on PFOODR to determine very accurately the potential energy function of the ground state via perturbed levels in the J G system. We are able to observe rotationally resolved levels up to 98

R3: The sensitivity of four-wave mixing (4WM) schemes relative to fluorescence- or ionization-detected incoherent double resonance (OODR) schemes has been discussed by Rohlifing and others. 4WM is background-free so it is more sensitive for strong transitions than the incoherent schemes, but the incoherent schemes are more sensitive for weak transitions because the signal dependences on

different powers of transition moments, which is of extreme importance for observation of weak extra lines at perturbations.

A3: We have added a two paragraphs in Section Isotope-Selective Tracing of Dark Bands:

The fast emission of a photon in a totally different spectral range gives these excitation scans a high sensitivity for detecting the dark states. While the non-linear four-wave mixing discloses the strongly perturbed lines of the dark states close to their intersection with the bright state, the linear incoherent method is more suitable for the observation of the weak transitions that are energetically more separated. TC-RFWM intensities depend quadratically on the population whereas laser-induced fluorescence intensities depend linearly which is advantageous in the low-density environment of a molecular beam. In addition, the non-linear method depends on different powers of the transition moment as the level of saturation changes[4] and is therefore further limiting the detection of weak transitions.

In Figures 3b and 3c numerous rotational levels of the dark $G_{62}0_u^+$ state are observed that are below the detection limit of four-wave mixing. The excitation scan in Figure 3b is obtained by monitoring the emission into the highly excited $v = 88$ level of the ground state $X0_g^+$ in the $^{65}\text{Cu}^{63}\text{Cu}$ isotopologue. The scan displays rotational levels of the $G_{62}0_u^+$ dark state up to $J' = 25$. The isotope shift of $v = 84$ in the ground state is not sufficiently large to separate the emission of the two main isotopologues by the limited resolution of the spectrometer. As a consequence, weak $^{63}\text{Cu}_2$ rotational P and R branch transitions up to $J' = 33$ of $G_{62}0_u^+$ are observed in addition as shown in the Figure 3c. In spite of the complex appearance of the excitation scans, the assignment of the dark states transitions is straightforward on the basis of the more intense transitions to the dark state that are unambiguously defined by double-resonance labeling.

R4: I suggest adding a paragraph about Spectroscopic diagnostics for multiple resonance schemes and for classes of perturbations. For example, polarization effects affect the intensity ratio for PR,PR and Q,Q transition sequences with parallel polarizations of the two lasers and PR,Q for perpendicular polarizations. Owing to the opposite signs of R vs. P transition amplitudes for perpendicular type ($=1$) transitions vs. same signs for parallel ($=0$) transitions, there will be anomalies in the R/P intensity ratio when there is an L-uncoupling ($=1$) [BJ.L] perturbation. These perturbations have matrix elements that are approximately linearly dependent on J, thus the size and even the sign of the R/P intensity anomaly becomes informatively J-dependent. There is a brief remark in the present text raising the possibility of such effects, but without any logical or useful structure.

A4: The paragraph has been re-written. The observation of the O1 perturbation in subsection Rotational disentanglement of bands in Results is followed by:

Intensity anomalies between $P(J+1)$ and $R(J-1)$ lines have been observed and attributed to the mixing of perturbing states.[1] The effect can be explained by quantum mechanical interference that comprises information on the perturbation class. For example, parallel and perpendicular transitions (with an orbital angular momentum change $\Delta\Lambda = 0$ or ± 1 , respectively) display approximately equal intensities for the $P(J+1)$ and $R(J-1)$ lines for unperturbed states. If a perturbation occurs (L-uncoupling) between two states exhibiting an angular momentum difference of $\Delta\Lambda = \pm 1$, the relative intensity of the branches out of this state can be affected strongly. For parallel transitions the amplitude phases of P and R are identical while for perpendicular transitions they are opposite. As a consequence, P line interference is constructive and the R line interferes destructively and the branch may disappear completely.[2] Numerous intensity anomalies occurring due to perturbation have been observed and their intensity patterns contain information on the class of perturbation (Ref. [2] and references therein). Additional information on perturbation effects is accessible by considering saturation and polarization features of the background-free double-resonant method applied in this work. In a simplified spectrum, exhibiting only few transitions owing to the stringent double-resonance selection rules, partially forbidden features (weak 'extra lines' obtaining oscillator strength through perturbation) can be observed at high laser intensities. Even though bright transitions might substantially broaden upon saturation they are in general well separated from the 'extra lines' that are made visible at increased laser powers. Furthermore, specific linear and circular polarization configurations of the two resonant

lasers allow further enhancement or discrimination of entire families of rovibronic transitions[3] for the characterization of perturbation effects. Considering the quantum mechanical interference for perturbed levels involving perpendicular transitions, l-uncoupling is suggested for the $O1 \sim G_{68}0_u^+$ system and consequently a 1_u symmetry label for the $O1$ state. However, the classification of this perturbations requires more detailed experiments and is beyond the scope of this report.

R5: What about nuclear permutation symmetry ortho:para intensity ratios for the 63-63 isotopologue and none for the 63-65?

A5: the intensity alterations in 63Cu2 have been observed and are documented (Visser, B., M. Beck, P. Bornhauser, G. Knopp, T. Gerber, R. Abela, J. A. van Bokhoven, and P. P. Radi. "Unraveling the Structure of Transition Metal Dimers Using Four-Wave Mixing." *Journal of Raman Spectroscopy* 47, no. 4 (April 2016): 425–31. <https://doi.org/10.1002/jrs.4841>.) For this investigation we think that a discussion of the nuclear spin statistics would be somewhat out of the scope. However, in the method section (H: Simulation of perturbed bands) we indicate: As the different statistical weight of symmetrical and asymmetrical rotational levels (3 and 5) in the homonuclear isotopologue is taken into account in the simulation, an average statistical weight of 4 was chosen for all levels of the heteronuclear isotopologue to match the average intensity.

R6: What about some discussion of what makes a state dark or bright with respect to a particular excitation path?

A6: In the abstract we added:

This allowed us to unwind the individual spectral lines by isotopic composition and rotational quantum number and revealed a rich network of bright states and perturbing dark ones, which gain their spectroscopic accessibility by quantum mechanical mixing with the dipole-allowed bright state. These mixed states serve as the starting point for transitions into otherwise inaccessible regions.

and in subsection spectral simplification by double-resonance spectroscopy (Section Results), 2nd paragraph:

Figure 1 shows an accurate simulation of the (0-0) J-X band and a Fortrat diagram, which maps the transitions to the individual rotational quantum number J' of the accessed state. There, not only transitions to the bright J state, but also hypothetical transitions into three dark vibronic states (which cannot be accessed by a single photon from the ground electronic state) are shown.

now reads:

Figure 1 shows an accurate simulation of the (0-0) $J0_u^+ - X0_g^+$ band and a Fortrat diagram, which maps the transitions to the individual rotational quantum number J' of the accessed state. There, not only transitions to the bright $J0_u^+$ state, but also hypothetical transitions into three dark vibronic states are shown. These states exhibit little transition strength from the initial ground state and cannot be accessed directly. However, they gain intensity and experience a line shift where their corresponding branches intersect with the $J0_u^+$ state.

R7: What about discussion of the calculable factors (vibrational overlap integral, interrelationships between spin-orbit and L-uncoupling matrix elements) that go into the value of a particular perturbation matrix element?

A7: We deal mainly with homogeneous perturbations between $0g+$ states. The classification of the G J perturbation has been added in the Method Section (C: Deperturbation and classification of the G67 state.

In addition, we have computed the overlap integrals and added a figure to the Extended Data and discussed the findings in the new Method Section F: Determination of the Franck-Condon overlaps in the Gxx J($v=0,1,2$) systems

Specific Details:

1. Page 2. Manifolddness is not an English word.

A: changed: it reads now:

In this regime, the partially filled d orbitals of transition metals lead to a very large number of electronic states.

2. Figure 1 caption and page 5. Define Fortrat parabola.

The Fortrat diagram is explained in the caption of Figure 1, last sentence:

Because dipole selection rules for this bands dictate that the rotational quantum number can change only by ± 1 for a given initial J'' , two branches occur for each state, a P and R-branch with $\Delta J = -1$ and $\Delta J = +1$, respectively.

3. Page 5. Detectable (not detectible).

A: corrected

4. Page 6. The state name G68 appears with little explanation.

A: we added a reference. The vide infra points to the details of the assignment in the methods section (i.e. C: Deperturbation and classification of the G67 state)

5. Page 6. "...swap their energy order..."

the sentence:

Most obvious is the strong interaction between the J ($v=2$) state and the vibronic state preliminarily labeled as G68. At the rotational levels $J = 29$ and $J = 30$, between which they swap their order, the repulsion of these levels increases their separation from 2 cm⁻¹ to 10 cm⁻¹.

now reads:

Most obvious is the strong interaction between the J ($v=2$) state and the vibronic state preliminarily labeled as G68. At the culmination, between rotational levels $J = 29$ and $J = 30$, the repulsion of these levels increases their separation from 2 cm⁻¹ to 10 cm⁻¹.

6. Page 7 "...beyond the known effects in diatomic molecules..." is a ridiculous and untrue statement, certainly not supported by reference 22.

7. Page 7. "...expected for a dark state..."

8. Page 7. Some needlessly vague and all-encapsulating statements about "intensity anomaly", "unexplained interference effects," and "polarization dependent studies".

9. Page 7. "...preliminarily..."

A to 6-9: We apologize for this ignorance. The paragraph has been completely rewritten. See above (A4).

10. Page 8. "Taking a look at the blue..." Blue in the spectrum or in the figure?

A: Taking a look at the blue fluorescence in Figures 3b and 3c, it becomes obvious why these features were missing.

now reads:

Taking a look at the blue fluorescence emitted around ≈ 452 and 446 nm in Figures 3b and 3c, respectively, it becomes obvious why these features were missing.

11. Page 9. "...increasing the number of samples averaged at each data point" What does this mean? the sentence is dropped. More information has been added that explains the sensitivity of FWM vs. non-coherent methods (see answer A3)

12. Page 9. the discussion of isotope shifts is clumsy and needlessly opaque.

the paragraph has been rewritten

13. Page 10-11. "...increases for higher vibrational excitation..." Is this a vibrational overlap or a J-dependent matrix element effect? A: As mentioned above, we have computed the overlap integrals that yield a good agreement with the experimental findings.

14. Page 11. The nature of the "anomaly" exhibited by the I state is unclear. Is the short lifetime due to predissociation or to an unexpectedly strong radiative transition?

A: As the other reviewer rationalized, the reason that the I state does not show perturbations might be due to the fact, that all observed perturbation are 0+g states. Since the I state is an 1u state, the heterogeneous perturbation would yield matrix elements roughly proportional to the rotational constant times J, times another constant that could be significantly less than 1. Such a perturbation might be too small to be observed.

The paragraph reads now:

The I_{1_u} state is an anomaly in many respects. Only a few vibrational levels have been observed and even these are too weak to be observed by most methods. We previously reported its short fluorescence lifetime and molecular constants of seemingly unperturbed regions.[5] The I_{1_u} state seems not to interact with the other states characterised in this work, but nevertheless is subject to a complicated pattern of perturbations. Therefore, the I_{1_u} state should be revisited in a subsequent study, so that the perturbing states may be investigated more thoroughly. Such a study can also help to assign the unrelated bands ($L1, M1, N1$ and $O1$) to specific electronic states. To be used for experimental purposes only, molecular constants that reproduce the observed range were added as Supplement Data Table 3.

15. Page 11. "Prepends" is not an English word. The sentence has been modified according to 14. above.

16. Page 12. I like "a glance behind the veil." But I do not like the double vagueness of "unexpected properties to be resolved by a new generation of methods."

A: The paragraph now reads:

Even for the copper dimer itself, a glance behind the veil of overlapping lines exposed not only a complex network of perturbations, but also vibronic bands displaying interesting properties to access regions of the energy map beyond the possibilities of more conventional spectroscopic methods.

Reviewer #3

Beck et al. present a spectroscopic study of electronic transitions of the Cu₂ dimer that are observed in the 37350-30850 cm⁻¹ range. They have used a two-dimensional technique to unravel the complex rotational band structures observed at these energies. This work is an extension of their previous 2-dimensional study of the same bands that was published in the Journal of Physical Chemistry A (reference 21). The present work reports additional measurements and a careful deperturbation analysis. While this is certainly a respectable study, the novelty is oversold. Many previous studies have shown the value of two-dimensional techniques for electronic spectroscopy, and the fact that Cu₂ has a high density of states above 30000cm⁻¹ is no surprise. I would not consider the intensity anomalies of transitions to the O1 state to be "beyond the known effects in diatomic molecules." In general, theoretical models of perturbation-induced intensity anomalies are well-developed. In summary, this is a quality spectroscopic study that provides new spectroscopic data for the Cu₂ dimer. I recommend publication after some softening of the claims of novelty.

A: We agree with the reviewer that the high density of states above 30000 cm⁻¹ is not surprising. Also, the merits of double-resonance techniques have been demonstrated in the literature. However, very little double-resonance experiments are available in the literature for transition metal dimers. In particular, two-color four-wave mixing on a transition metal is applied in our laboratory for the first time. Difficult experimental issues (like a stable source and the optical setup) have been solved to apply this non-linear method to a transition metal dimer in a molecular beam. The method yields a novel, much detailed view of the electronic structure of the dicopper molecule and opens ways to investigate other transition metal dimers and larger clusters in a similar fashion.

The intensity anomaly of the O1 state is discussed more accurately. The observation of the O1 perturbation in subsection Rotational disentanglement of bands in Results is followed by

Intensity anomalies between $P(J+1)$ and $R(J-1)$ lines have been observed and attributed to the mixing of perturbing states.[1] The effect can be explained by quantum mechanical interference that comprises information on the perturbation class. For example, parallel and perpendicular transitions (with an orbital angular momentum change $\Delta\Lambda = 0$ or ± 1 , respectively) display approximately equal intensities for the $P(J+1)$ and $R(J-1)$ lines for unperturbed states. If a perturbation occurs (L -uncoupling) between two states exhibiting an angular momentum difference of $\Delta\Lambda = \pm 1$, the relative intensity of the branches out of this state can be affected strongly. For parallel transitions the amplitude phases of P and R are identical while for perpendicular transitions they are opposite. As a consequence, P line interference is constructive and the R line interferes destructively and the branch may disappear completely.[2] Numerous intensity anomalies occurring due to perturbation have been observed

and their intensity patterns contain information on the class of perturbation (Ref. [2] and references therein). Additional information on perturbation effects is accessible by considering saturation and polarization features of the background-free double-resonant method applied in this work. In a simplified spectrum, exhibiting only few transitions owing to the stringent double-resonance selection rules, partially forbidden features (weak 'extra lines' obtaining oscillator strength through perturbation) can be observed at high laser intensities. Even though bright transitions might substantially broaden upon saturation they are in general well separated from the 'extra lines' that are made visible at increased laser powers. Furthermore, specific linear and circular polarization configurations of the two resonant lasers allow further enhancement or discrimination of entire families of rovibronic transitions[3] for the characterization of perturbation effects. Considering the quantum mechanical interference for perturbed levels involving perpendicular transitions, l-uncoupling is suggested for the $O1 \sim G_{68}0_u^+$ system and consequently a 1_u symmetry label for the $O1$ state. However, the classification of this perturbations requires more detailed experiments and is beyond the scope of this report.

References

- [1] Ikoma, H., Kasahara, S. & Katô, H. Perturbations, intensity anomalies, and line broadening of $^{23}\text{Na}^{39}\text{K}$ studied by optical-optical double resonance polarization spectroscopy. *Molecular Physics* **85**, 799–820 (1995).
- [2] Lefebvre-Brion, H. & Field, R. W. Chapter 2 - Basic Models. In Field, H. L.-B. W. (ed.) *The Spectra and Dynamics of Diatomic Molecules*, 61–86 (Academic Press, San Diego, 2004).
- [3] Murdock, D., Burns, L. A. & Vaccaro, P. H. Dissection of Rovibronic Structure by Polarization-Resolved Two-Color Resonant Four-Wave Mixing Spectroscopy†. *The Journal of Physical Chemistry A* **113**, 13184–13198 (2009).
- [4] Farrow, R. L., Rakestraw, D. J. & Dreier, T. Investigation of the Dependence of Degenerate 4-Wave-Mixing Line-Intensities on Transition Dipole-Moment. *Journal of the Optical Society of America B-Optical Physics* **9**, 1770–1777 (1992).
- [5] Beck, M. *et al.* Rovibrational Characterization of High-Lying Electronic States of Cu_2 by Double-Resonant Nonlinear Spectroscopy. *J. Phys. Chem. A* **121**, 8448–8452 (2017).

REVIEWERS' COMMENTS:

Reviewer #1 (Remarks to the Author):

This revised manuscript has been extensively reworked to improve its clarity and represents a major improvement over the originally submitted version. It represents very high quality spectroscopic work and is certainly worthy of publication. I have a few remaining minor suggestions that should be considered.

1. On page 2, first column, about 60% down the column, the authors state "...thus determining the complete configuration of the $J^0_{u^+}$ state as $3d^{18}3d\pi_g 4s\sigma_g^2 4p\pi_u$, a state that corresponds to the $^2S + ^2P$ asymptotic limit." I think this statement is a bit confusing because the only 2P state the authors could be referring to corresponds to excitation of the 4s electron to the 4p orbital and therefore retains a completely filled 3d orbital. This is not consistent with the excitation of a 3d electron to the 4p level, which makes the statement as written slightly confusing. I also believe that it is important to designate the parity of atomic states, which could be done either by referring to the separated atom limit as the $^2S_g + ^2P_u$ limit (my preference) or as the $^2S + ^2P^o$ limit, as is done at the NIST atomic spectra site. I would recommend revising the sentence to read something like this: "...thus determining the complete configuration of the $J^0_{u^+}$ state as $3d^{18}3d\pi_g 4s\sigma_g^2 4p\pi_u$, a state that dissociates adiabatically to the $^2S_g + ^2P_u$ asymptotic limit."

2. On page 2, line 4 of paragraph 2 of section IIA, I think "hypothetical" should be changed to "hypothesized". Hypothetical suggests something that might be contrary to fact, while hypothesized suggests a fact that the authors are suggesting to be true. This is a very minor point.

3. In several places in the manuscript, the authors refer to the O_1 state and on page 4 they suggest that this is a state that heterogeneously couples to the G_{680}^+ state, and must have $\Omega = 1$. If this is the assignment, I think it would be useful to refer to the state as the O_{1u} state throughout the paper. Similarly, the authors refer to the L1, M1, and N1 bands but it is not clear whether they believe these correspond to $\Omega=1$. If so, then the states must have $\Omega = 1$ and they should be labeled as L_{1u} , M_{1u} , and N_{1u} .

4. Near the bottom of the first column on page 4, the authors use “preliminary” as an adverb. This should be changed to “preliminarily”.

5. On page 8, column 2, line 3, change “is certainly a $\Omega = 0$ state” to “is certainly an $\Omega = 0$ state” Likewise, in lines 11 and 12 from the bottom, change the text to read “...only two covalently bound 0 states interact with the 0 ion pair ground state...”. Also, line 6 from the bottom, change “further 0 states” to “further 0 states”

Overall Recommendation:

With these minor changes, the manuscript will be acceptable for publication. No further review is required.

Reviewer #2 (Remarks to the Author):

I am satisfied by the author's responses to all of my suggestions and questions. This is a magnificent paper, ideally suited for publication as a Nature Communication.

Robert W. Field

Reviewer #3 (Remarks to the Author):

The authors have made sufficient changes in response to the comments of the reviewers. The paper is now suitable for publication.

We appreciate very much the comments to our revised paper from Reviewer I. We address the issues detailed in the following. Changes are indicated in blue color:

This revised manuscript has been extensively reworked to improve its clarity and represents a major improvement over the originally submitted version. It represents very high quality spectroscopic work and is certainly worthy of publication.

I have a few remaining minor suggestions that should be considered.

1. On page 2, first column, about 60% down the column, the authors state "...thus determining the complete configuration of the $J0_u^+$ state as $3d^{18}3d\pi_g4s\sigma_g^24p\pi_u$, a state that corresponds to the $^2S + ^2P$ asymptotic limit." I think this statement is a bit confusing because the only 2P state the authors could be referring to corresponds to excitation of the 4s electron to the 4p orbital and therefore retains a completely filled 3d orbital. This is not consistent with the excitation of a 3d electron to the 4p level, which makes the statement as written slightly confusing. I also believe that it is important to designate the parity of atomic states, which could be done either by referring to the separated atom limit as the $^2S_g + ^2P_u$ limit (my preference) or as the $^2S + ^2P^o$ limit, as is done at the NIST atomic spectra site. I would recommend revising the sentence to read something like this: "...thus determining the complete configuration of the $J0_u^+$ state as $3d^{18}3d\pi_g4s\sigma_g^24p\pi_u$, a state that dissociates adiabatically to the $^2S_g + ^2P_u$ asymptotic limit."

A1: The reviewer is correct: The asymptotic limit we are referring to is incorrect. We have corrected the sentence as follows:

"...thus determining the complete configuration of the $J0_u^+$ state as $3d^{18}3d\pi_g4s\sigma_g^24p\pi_u$, a state that dissociates adiabatically to the $3d^94s4p + 3d^{10}4s$ asymptotic limit."

2. On page 2, line 4 of paragraph 2 of section IIA, I think "hypothetical" should be changed to "hypothesized". Hypothetical suggests something that might be contrary to fact, while hypothesized suggests a fact that the authors are suggesting to be true. This is a very minor point.

A2: We have modified the word according to the reviewers' suggestion.

3. In several places in the manuscript, the authors refer to the O1 state and on page 4 they suggest that this is a state that heterogeneously couples to the G680u+ state, and must have $\text{OMEGA} = 1u$. If this is the assignment, I think it would be useful to refer to the state as the O1u state throughout the paper. Similarly, the authors refer to the L1, M1, and N1 bands but it is not clear whether they believe these correspond to $\text{OMEGA}=1$. If so, then the states must have $\text{OMEGA} = 1u$ and they should be labeled as L1u, M1u, and N1u.

In fact, the O1 state seems to be an $\Omega = 1_u$ state. However, to assign this state unambiguously, further experiments are required as stated at the end of Section II B:

"However, the classification of these perturbations requires more detailed experiments and is beyond the scope of this report."

We suggest that prior to a definitive determination the Ω -label should be omitted. In a similar way, the L1, M1 and N1 need further investigations for an accurate assignment. There is a statement at the end of Section II D, which clarifies the consideration:

"The $I1_u$ state seems not to interact with the other states characterised in this work, but nevertheless is subject to a complicated pattern of perturbations. Therefore, the $I1_u$ state should be revisited in a subsequent study, so that the perturbing states may be investigated more thoroughly. Such a study can also help to assign the unrelated bands (L1, M1, N1 and O1) to specific electronic states."

4. Near the bottom of the first column on page 4, the authors use "preliminary" as an adverb. This should be changed to "preliminarily".

A4: The sentence has been corrected:

For the copper dimer, a good example is provided by the series of vibrational levels that were preliminarily assigned to the $G0_u^+$ state.

5. On page 8, column 2, line 3, change “is certainly a $= 0$ state” to “is certainly an $= 0u+$ state” Likewise, in lines 11 and 12 from the bottom, change the text to read “. . . only two covalently bound $0u+$ states interact with the $0u+$ ion pair ground state. . .”. Also, line 6 from the bottom, change “further $0+$ states” to “further $0u+$ states”

A5: We have modified all affected term symbols accordingly

Overall Recommendation: With these minor changes, the manuscript will be acceptable for publication. No further review is required.